# Characterization of Dissolved Organic Matter in Solar Ponds by Elemental Analysis, Infrared Spectroscopy, Nuclear Magnetic Resonance and Pyrolysis–GC–MS

**DOI:** 10.3390/ijerph19159067

**Published:** 2022-07-25

**Authors:** Keli Yang, Yaoling Zhang, Yaping Dong, Jiaoyu Peng, Wu Li, Haining Liu

**Affiliations:** 1Key Laboratory of Comprehensive and Highly Efficient Utilization of Salt Lake Resources, Qinghai Institute of Salt Lakes, Chinese Academy of Sciences, Xining 810008, China; zhyl@isl.ac.cn (Y.Z.); dongyaping@hotmail.com (Y.D.); pengjy@isl.ac.cn (J.P.); driverlaoli@163.com (W.L.); liuhn@isl.ac.cn (H.L.); 2Qinghai Technology Research and Development Center of Comprehensive Utilization of Salt Lakes Resources, Xining 810008, China; 3Qinghai Provincial Key Laboratory of Resources and Chemistry of Salt Lakes, Xining 810008, China

**Keywords:** chemical composition, Salt Lake brine, evaporation, mineral extraction, transformations

## Abstract

The abundance and chemical composition of dissolved organic matter (DOM) in the brine of solar ponds affect the efficiency of mineral extraction and evaporation rates of the brine, and cause undesired odor and color of the products. Here, we report an investigation into the composition and changes of DOM in solar ponds from Salt Lake brine with multiple complementary analysis techniques. The results showed that the DOM derived from Salt Lake brine was primarily composed of carbohydrates, aliphatic and aromatic compounds. The concentrations of dissolved organic carbon in solar ponds increased with exposure time by up to 15−fold (from 23.4 to 330.8 mg/L) upon evaporation/irradiation of Salt Lake brine. Further qualitative analyses suggest that the relative abundance of aliphatic compounds (including functionalized ones) increased from 49.5% to 59.2% in the solar pond process, while the opposite was observed for carboxylic acid moieties, aromatics and carbohydrates, which decreased from 15.7%, 7.1% and 26.1% to 13.4%, 5.3% and 23.0%, respectively. The pyrolysis–gas chromatography–mass spectrometry results reveal that the presence of some sulfur-containing organics implied some anaerobic biotic decay, but microbiological processes were probably subordinate to photo-induced DOM transformations. In the Salt Lake brine, exposure-driven decay decreased the abundance of polysaccharides and increased that of mono- and polyaromatic pyrolysis products. Our results here provide new insights for better understanding the changes of DOM chemical composition in the solar ponds of Salt Lake brine.

## 1. Introduction

Salt Lake brine usually contains raw materials such as potassium chloride, lithium carbonate, boron, sodium and magnesium for mineral extraction [1,2,3]. Due to technical and economic viability, the solar pond has been used successfully in mineral extraction from Salt Lake brine [4,5]. Dissolved organic matter (DOM), the most active fractions in solar ponds, often have adverse effects on the mineral extraction process related to the evaporation rate and degree, and further affect the quality of the products (e.g., smell, color and purity), especially for the highly valuable or hyper-pure products, due to its adsorption or binding behaviors with inorganic components [6,7,8]. Thus, it is necessary to understand the chemical structures of brine DOM, so as to develop effective treatment strategies for DOM removal.

Various analysis tools for DOM characterization have been conducted in the last decades [9,10,11,12,13,14]. However, there is no single analytical method that can reveal the actual chemical composition of DOM in aqueous systems because of the extreme heterogeneity and complexity of DOM, and multiple analysis methods are usually required [15]. For instance, both UV-visible absorption and fluorescence can only provide information on chromophoric DOM (or CDOM), rather than the total DOM pools [16,17]; elemental analysis, Fourier transform infrared spectroscopy (FTIR) and nuclear magnetic resonance (NMR) can only reveal the composition of bulk or functional groups of DOM [18,19,20,21,22]. For example, elemental analysis can provide the DOM elemental composition (C, H, O, N, S), while FTIR can further give the functional groups of DOM. In addition, NMR not only provides more detailed functional structures, but also provides their quantitative information. Recent studies have shown that the technique of pyrolysis–gas chromatography–mass spectrometry (Py–GC–MS) has extended the characterization level to the molecular scale, which can break macromolecular DOM into fragments and enable rapid screening of macromolecular constituents of DOM, such as proteins, lignin, carbohydrates and so on [23,24,25,26]. However, this technique is powerless when it probes into DOM with polar functional groups (-NH, -OH, -COOH) typically presenting in aquatic DOM [23]. Fourier Transform Ion Cyclotron Resonance Mass Spectrometry (FT-ICR-MS) is characterized by a high degree of mass accuracy and precision, and can give distributions of elemental formulae [27,28]. However, these studies are limited to high-purity DOM fractions and by expensive costs. Thus, complementary chemical characterization techniques are essential for a comprehensive understanding of the chemical composition of DOM. Moreover, the abundance and composition of DOM undergoes significant changes, such as photodegradation and evaporation loss, due to intense sunlight exposure and high temperatures in solar ponds, which further affect the degree of evaporation and the quality of the products extracted from the brine. The investigation into the changes in DOM composition in solar ponds provides a baseline dataset from which to understand future changes during exploitation of Salt Lake resources.

In this study, the changes of chemical composition of DOM derived from solar ponds were investigated using Da Qaidam Salt Lake brine, located in the Qaidam basin, Qinghai–Tibet Plateau region. Solid-phase extraction with PPL cartridges, which reportedly produce high DOC recovery and representative DOM fractions [29,30,31], was used to isolate DOM from the hypersaline water. The DOM was characterized by elemental composition, FTIR, ^13^C NMR and Py–GC–MS. This work complements and extends our knowledge on the chemical structures and changes of DOM in solar ponds, which is essential to guide the development of environmental management and treatment strategies for high-value productions extracted from Salt Lakes.

## 2. Materials and Methods

### 2.1. Sites Description and Sampling

Da Qaidam Salt Lake, is located in the north of the Qaidam basin in Qinghai and is surrounded by the Luliang Mountains and Daken Daban Mountains (Figure 1), belonging to the Qilian Mountain range, and was formed during the Pleistocene period [32]. It has an area of 240 km^2^, the maximum width extending ~3.9 km and length ~15.8 km, and the depth ranges from 0.2 to 0.7 m. The brine of the lake contains huge amounts of useful mineral elements (e.g., potassium, lithium, boron, sodium and magnesium), which has been well described in our previous studies [30]. Due to the shortage of fossil fuels in this remote area, solar pond technology is the main solution to separate and extract minerals from brine. The industrial mineral exploitation in Da Qaidam Salt Lake is mainly conducted by the mining company Dahua Industry Company Ltd., which built a production plant and has produced potassium chloride, potassium sulfate and lithium chloride since 2003.

The solar ponds were constructed near the Salt Lake (Da Qaidam Salt Lake, 37°50′50″ N, 95°14′42″ E). The solar pond of Da Qaidam Salt Lake (Figure 2), used to obtain potash primary products, includes the initial brine pond (SIP), the sodium pond (SSP), the regulation pond (SRP), the carnallite pond (SCP, the main pond for primary commodity extraction) and the bittern pond (SBP). The brine samples from different stages of the solar ponds were collected in two periods (2017 and 2018). Before collection, each of the acid-cleaned carboys were thoroughly rinsed with the sample. Each sample was filtered immediately with 0.7 μm GF/F glass fiber (Whatman, pre-combusted at 450 °C for 5 h) in the field. Subsequently, the <0.7 μm samples were covered with ice packs in the dark, and transported to the lab for DOM isolation and purification as soon as possible.

### 2.2. Physical and Chemical Characterization

The dissolved fractions of total, inorganic and organic carbon concentrations of the brine samples were determined using a total organic carbon analyzer (Analytik Jena multi N/C 3100, Thuringia, Germany) with a high-temperature catalytic oxidation method. Each sample was diluted (1:10) with ultrapure water prior to analysis due to hypersaline characteristics. All the values were obtained with variation coefficients less than 3.0%, and the detection limits were 0.1 ppm. The contents of major cations and anions, such as Na^+^, Mg^2+^ and Cl^−^, were determined by traditional titration methods with the average analytical error of triplicate titrations being less than 0.3%. The concentration of Li^+^ was analyzed by inductively coupled plasma optical emission spectroscopy (ICAP 6500 Duo, Thermo Scientific, Waltham, MA, USA), with detection limits less than 0.1 mg/L and the standard deviations of the methods less than 6.0%. All the mentioned data were obtained by three analytical replicates.

### 2.3. Isolation and Purification of DOM

The DOM samples were isolated and purified following a procedure established in our previous studies [29,30]. Briefly, the PPL cartridges (Agilent Technologies, Santa Clara, CA, USA) were activated with 10 mL ethanol. Next, 500 mL acidified samples (pH 2) were passed through the cartridges using a pump (Gast Company, Benton Harbor, MI, USA) at a flow rate of 3 mL/min. Before elution, the cartridges were rinsed with 20 mL 0.01 mol/L HCl to remove excess salts. The DOM were eluted with 10 mL methanol, and were dried and stored in a desiccator for further analysis. Blank controls were conducted with Milli-Q water acidified with HCl (pH 2) using the same procedures.

### 2.4. Elemental Analysis

The relative contents of C, H and N of the purified DOM samples were investigated in duplicate using elemental analyzer equipment (EL CUBE, Heraeus, Germany). The percentages of oxygen were calculated by subtracting the relative contents of C, H and N from 100%, and trace fractions of S and P were ignored. The carbon stable isotopic determination (δ^13^C) was determined using a Delta plus XP isotope ratio mass spectrometer (Thermo Finnigan, San Jose, CA, USA), and its equation as follows:δ^13^C value (‰) = [(*R*_sample_/*R*_standard_)−1] × 100%(1)
in which *R* is the ratio of ^13^C/^12^C.

### 2.5. FTIR Analysis

The spectra of FTIR were obtained using a FTIR spectrometer (Thermo Nicolet NEXUS 670, Waltham, MA, USA) from 400 to 4000 cm^–1^ with 64 scans. Pellets were prepared by pressing a 2 mg sample and 200 mg KBr (spectroscopy grade, 1:100) under vacuum. Prior to analysis, the FTIR spectra and second-derivative spectra were normalized, and the reconstructed data matrix was then progressed using the drawing software.

### 2.6. Solid-State ^13^C NMR Analysis

Cross polarization/magic angle spinning solid (CP/MAS) nuclear magnetic resonance (^13^C NMR) spectroscopy was recorded by a Bruker Advance DSX 300 MHz with 50 mg samples from redissolved solids in CD_3_OD with Bruker standard pulse sequences and 5 mm cryogenic probes, and a rotor spin speed of 4500 Hz; the optimal relaxation delay was 3 s, and the contact time was 2 ms.

### 2.7. Pyrolysis (Py−GC−MS) Analysis

The pyrolysis of 2 mg DOM samples was conducted on a pyrolysis instrument (Py, CDS Pyroprobe 2000, Rochester, NY, USA), which heated at a rate of 5 °C/ms from 250 to 610 °C and then remained at 610 °C for 10 s. Pyrolysis products were separated and detected on a GC–MS (QP2010, Shimadzu Corporation, Kyoto, Japan). The split mode was 1:40; the helium carrier gas speed was 1 mL/min; and the capillary column was HP-5MS (length 30 m × thickness 0.25 mm × diameter 0.25 μm). The MS detector was selected at 70 eV electron ionization, and the *m*/*z* scan ranged from 45 to 650.

## 3. Results and Discussion

### 3.1. Physical and Chemical Characterization

The pH, total dissolved solid (TDS) and major ion contents of the five samples are listed in Table 1. The pH was slightly basic in the initial stages of the solar ponds (7.4 in SIP and 7.2 in SSP) and mildly acidic in the late stages of the solar ponds (6.6 in SRP, 4.6 in SCP and 4.6 in SBP). The contents of TDS ranged from 273.5 to 479.6 g/L, which was 1.7 times the chemical evolution of brine along the natural evaporation with solar pond technology. The dominant inorganic constituents were Li^+^, Na^+^, Mg^2+^, K^+^, Cl^−^, B_2_O_3_ and SO_4_^2−^ in the Salt Lake brine. The pattern of the inorganic constituents was similar to that of water produced by shale gas and tight gas sands from certain oil wells in the central and western U.S., which are characterized by high magnesium concentration [33]. In different evaporation stages, the varied trends were different. The concentration of Cl^−^ increased from 107.8 to 277.1 g/L, B_2_O_3_ increased from 0.6 to 40.1 g/L, Mg^2+^ increased from 15.0 to 114.9 g/L, and Li^+^ increased from 0.1 to 3.0 g/L in the solar ponds. However, the reverse results were true for Na^+^ concentrations, which decreased from 119.8 to 25.5 g/L, as most of the Na^+^ was crystallized out. Moreover, the concentrations of SO_4_^2−^ and K^+^ exhibited an initial increase followed by a decline. The variation trends of inorganic constituent concentrations can be attributed to the different evolution courses of inorganic ions along the natural evaporation of brine [2].

The contents of dissolved total, inorganic, organic carbon and total nitrogen (TN) are shown in Table 1. The dissolved inorganic carbon (DIC) concentrations have no systematic trend, ranging from 8.3 (SCP) to 40.9 g/L (SSP). The DOC and TDS obviously increased (*r* = 0.97, *p* < 0.01) with increasing exposure time. This implies that the DOM loss by degradation was compensated by an increased concentration of DOM due to evaporation in the solar ponds and/or microbial growth. The DOC and total TN contents were strongly correlated (*r* = 0.99), suggesting that the majority of the N existed in an organic form.

### 3.2. Elemental Analysis

The values of elemental compositions, atomic ratios (H/C, O/C and N/C) and δ^13^C of the purified DOM samples derived from the solar ponds in this study are listed in Table 2. The relative contents of C, H, N and O accounted for 37.1–53.5%, 3.8–5.2%, 1.9–2.4% and 39.2–56.2% of the total mass, respectively. In general, these values were in the same ranges as reported in the literature [29,34,35], except that the oxygen contents of SRP DOM were slightly high and the H contents decreased.

To further investigate the structural changes of DOM samples in the solar ponds, the atomic ratios of H/C, C/N and O/C were calculated based on the analysis results. The H/C ratio is an indicator of the degree of maturity or aromatic condensation of DOM, and that of C/N and O/C can indicate the contents of N-containing compounds and carbohydrate/carboxylic acid in DOM samples, respectively [35]. As depicted in Table 2, the DOM samples in initial stages of the solar ponds generally contained lower values of H/C and C/N than that of late stages of the solar ponds, while O/C ratios exhibited an opposite trend. The results indicate that the contents of saturated aliphatic increased, but N/O-containing compounds decreased with increasing exposure time in the solar ponds. This can also be confirmed by the results of FTIR and ^13^C CP/MAS NMR analysis. These phenomena suggest transformations of unsaturated hydrocarbons to saturated aliphatic compounds in solar ponds. In addition, the δ^13^C of all DOM samples were relatively constant, ranging from −21.0 to −22.2%, indicating a significant in situ origins for this material [36,37].

### 3.3. FTIR Characterization of DOM

As depicted in Figure 3a, the major peaks were located around 3450 cm^−1^, 3200 cm^−1^, 2920 cm^−1^, 1720 cm^−1^, 1640 cm^−1^, 1410 cm^−1^ and 1050 cm^−1^. The band around 3400 cm^−1^ is ascribed to the O-H bonds in water and carbohydrates, and intra- and inter-molecular hydrogen bonds in compounds such as phenols, carbohydrates and carboxylic acids [38]. The peak at 3230 cm^−1^ could be assigned to the stretching of N-H [10]. The intensities of these peaks significantly decreased from SIP DOM to SBP DOM, indicating that O-H/N-H-containing groups decreased in the late stages of the solar ponds.

The adsorption peaks at approximately 2970−2920 cm^−1^ are ascribed to C-H stretching of aliphatic CH_3_ or CH_2_ [29]. The peak areas of these DOM samples (except the SSP DOM) for the region (2970−2920 cm^−1^) increased with increasing exposure time (from 15.9% to 20.7%), which implies that aliphatic compounds increase in the late stage of the solar pond process. The trend could be partly attributed to photochemical oxidation of aromatic portions resulting in the DOM transformation, which leaves the more aliphatic groups [39].

The adsorption bands at 1720 cm^−1^ and 1410 cm^−1^ indicated the presence of protonated/de-protonated carboxylic acid functional groups [15]. The peaks around the center of 1640 cm^−1^ are assigned to the C=O stretching in amide I and the vibration of N-H or C-C stretching of aromatic rings. A weak band in the 1050 cm^−1^ is attributed to C-O stretching of carbohydrates [40]. As depicted in Figure 3b, these peak intensities decreased significantly from SIP DOM to SBP DOM. Moreover, the second derivative spectra suggest that both the band of N-H/C-N vibration in amide II (1540 cm^−1^) and the band of aromatic ring vibration (1515 cm^−1^) also exhibited the decreasing trend. These results indicate that the relative contents of carboxylic acids, aromatics, carbohydrates and nitrogen-containing compounds decrease in the solar ponds, which conform to the results of the elemental analysis.

#### 3.4. ^13^ C CP/MAS NMR Spectroscopy

In general, the ^13^C NMR spectra (Figure 4) can be divided into seven main regions, i.e., 0–45 ppm (alkyl C), 45–65 ppm (methoxyl/N-alkyl C), 65–110 ppm (O-alkyl C), 110–145 ppm (aromatic C), 145−160 ppm (aromatic C-O), 160–190 ppm (carboxyl/amide C) and 190–220 ppm (carbonyl C). The corresponding normalized distributions of these regions were calculated using the instrument software (Table 3).

For the alkyl C region, the central peaks at 24 ppm and 44 ppm could be attributed to CH_3_ in branched and short-chain compounds, and the CH_2_ in long-chain structures, respectively [35,41]. The methoxyl/N-alkyl C signal corresponds to ether, alcohol and the other hetero-aliphatic compounds [42]. The location at 56 ppm attributed to the methoxyl carbons attached to the syringyl or guaiacyl in the lignin compound was absent. This confirmed a previous study that there was virtually no or minor terrestrial-derived DOM input (vascular plants) in the Salt Lake [29]. The O-alkyl C region was dominated by carbohydrates, including the well-resolved peak at 85 ppm from carbohydrate ring carbon atoms, while the relatively weak signal at 108 ppm was associated with anomeric carbons [43]. For the 110−145 ppm region, the location at 132 ppm is related to benzene ring carbons, accounted for 4.1–5.4%, which was generally lower than that in an estuary on the eastern coast of Hainan Island (3.97–15.96%) [44]. While the aromatic C-O is usually ascribed to phenolic carbon. In the carboxyl/amide carbon region, the central peak at 176 ppm could be ascribed to amide, carboxyl, as well as ester carbons. The 190 to 220 ppm region could be attributed to aldehyde and ketonic carbons.

The spectra of all DOM samples were similar, suggesting that the DOM isolated from different solar ponds contained similar group distributions. Among them, the aliphatic, carbohydrate and aromatic carbons accounted for 49.5−59.3%, 21.7−26.2% and 5.3−7.1% of the total signal, respectively (Table 3). In general, the aliphatic C values and methoxyl/N-alkyl C showed an increasing trend in the solar ponds during an evaporation/irradiation treatment (*r* = 0.91; *p* < 0.001), while the O-alkyl C (*r* = −0.84, *p* < 0.001), total aromatics (*r* = −0.79; *p* < 0.05) and carboxyl/amide C (*r* = −0.77; *p* < 0.001) exhibited an opposite trend. The results agree with the observation of the elemental composition and FTIR analysis. In general, the components with functional groups were more likely to decay or transform compared to saturated hydrocarbons in the solar pond. Thus, aliphatics may be the target compounds to remove or treat for the extraction of high-value Salt Lake productions.

#### 3.5. Changes in the Molecular Composition of the DOM Based on Py−GC−MS Analysis

The DOM samples contained prolific compounds with numerous isomers and side-chain configurations, including alkane/alkene pairs, alkyl-substituted monoaromatic and polyaromatic hydrocarbons (MAHs/PAHs) and alicyclic compounds (Appendix A). The products were grouped into 11 subclasses based on chemical structures (Table 4). Remaining compounds, including unidentified ones, were labelled as “other compounds”. The total ion current pyrograms of the Py-GC/MS analysis for the DOM samples is shown in Appendix A.

On the whole, the most abundant group (24.99−28.08%) was 1-ring unsaturated alicyclic compounds (ALICYCL) based on cyclopentene and cyclohex(adi)enes groups, such as cyclohexadiene, C_1_-cyclohexene, C_1_-cyclohexadienes (4 isomers), C_2_-cyclohexadienes (10 isomers), C_3_-cyclohexadienes (7 isomers) and C_4_-cyclohexene (limonene). They represented a cyclic aliphatic component of the DOM. They may be derived from terpenoids and have similar features to the DOM from the Great Salt Lake [45]. It is argued that the terpenoid-like DOM was probably degradation products of algal and bacterial precursors (pigments and steroids therein). Poorly identified terpenoid-like structures were often found in the DOM samples, sometimes containing abundant carboxylic groups. Such carboxylic groups could be present in the source of the alicyclic compounds, and were eliminated during analytical pyrolysis (decarboxylation).

The second most abundant group (19.92−27.59%) had structures that is typical of the pyrolysis products of carbohydrates (CARB), such as acetic acid, hydropropanone, hydroxypropanal, cyclopentenones, furans and furaldehydes. These compounds probably originated from microbial polysaccharides, or other kinds of carbohydrates after significant influences of (photo-)degradation. Pyrans and anhydrosugars such as levoglucosan were not detected, suggesting that vascular plant-derived carbohydrates (holocellulose derivatives) were negligible. This is in agreement with the negligible fluxes of rivers and streams in the xeric environments of the Qaidam Basin [29]. The carbohydrate products probably reflected a labile component of algae and bacteria. The decreasing trends of CARB exposure to sunlight in solar ponds further enhanced this conclusion. Several Salt Lake DOM samples from North America also showed a significant carbohydrate component [46].

Monocyclic aromatic hydrocarbons (MAHs) included benzene and alkylbenzenes with 1 (toluene) to 5 carbon atoms in alkyl groups. This series included mono-unsaturated MAHs such as styrene, and accounted for 16.79−22.54%. The MAHs were detected in most DOM studies by Py–GC–MS or GC–MS, from (hyper) saline environments. Benzene, toluene, xylenes and other MAHs were also detected in oilfield-produced brine studies using GC–MS of dichloromethane-soluble DOM, indicating that the MAHs were not secondary pyrolysis products of non-aromatic DOM [47]. In previous studies, the aromatics were confirmed to be of the preferential photo-degradation of DOM components [48,49]. The increasing trend of MAHs here may be related to the decomposition of the polyphenolic and condensed aromatic compounds, which released more low molecular weight MAHs [50].

The (linear) n-alkanes, n-alkenes, isoprenoid alkanes and alkenes with a chain length ranging from C_6_ to C_23_ accounted for 11.21−15.69%. They were grouped as methylene chain compounds (MCC). Other MCC such as fatty acids, methylketones, alkylnitriles or alkylamides were not detected. These patterns were probably related to aliphatic materials in microbial sources. The lack of phytadienes excluded a significant source in fresh phytoplankton, but intense photo-oxidation could efficiently eliminate such moieties. The isoprenoid compounds included C_9_-alkadiene (diemethylheptadiene compound), a C_12_-isoprenoid alkanone and the other unidentified isoprenoid ketones. These were uncommon pyrolysis products and highlighted the idiosyncratic nature of the DOM in hypersaline water. They might originate either from a specific kind of diatom or bacterium, or they may be oxidation products of any isoprenoid MCC precursor.

The detected phenols (PHEN), including phenol, methylphenols, C_2_-alkylphenols (4 isomers) and a C_3_-alkylphenol, accounted for 3.28−3.88%. These compounds could originate from a variety of sources, such as microbial ones. A partial confusion with C_9_-isoprenoid alkatriene groups (m/z 107 and 122) for the dimethylphenols could not be excluded.

The sum of compounds with a halogen atom (HALOGENCOMP) in its structure (0.47−1.65%) was detected in our study. These compounds included bromomethane (MeBr) and iodomethane (MeI), but other organohalogens could be among the unidentified products. Polycyclic aromatic hydrocarbons (PAHs) were mostly indenes (C_0_-C_3_-alkylindenes) and naphthalenes (C_0_-C_2_). The extensive methylation probably indicated a terpenoid-like or asphaltene source, even though contamination from samples treatment cannot be excluded.

Compounds with nitrogen (NCOMP) in our study were scarce (1.43−1.84%) and dominated by pyrroles (N-methylpyrrole, 2-methylpyrrole and 3-methylpyrrole), with traces of benzonitrile and C_3_/C_4_-alkylanilines. These compounds should probably be ascribed to microbial DOM. The lack of indoles, acetamides, cyanobenzenes and diketopiperazines indicated that the N-rich DOM was strongly affected by decay processes, as intact proteins, chitins or peptidoglycans would generate a more diverse N product fingerprint upon Py−GC−MS. This decay process could also explain the low abundance of organic N; although a prevailing microbial source of the DOM, N-rich biopolymers tended to be relatively labile components. The only sulfur-containing products (SCOMP) were benzothiazole, S_2_ and/or SO_2_ (0.66−2.19%). The identification of thiophene was tentative and was added to the “other compounds”.

A peak with m/z 81, 109 and 124 at the expected retention time of guaiacol was the only possible sign of lignin (LIG), which is an important marker of vascular plants, mainly deriving from terrigenous sources. The low percentages (0.30–0.43%) indicated autochthonous characteristics of these DOM. According to carbon stable isotopic analysis, these materials significantly exhibited in situ origins, and our results here appear to reinforce this viewpoint. Finally, the other compounds (4.89−6.20%) included numerous unidentified peaks, peaks from plasticizers (phthalic anhydride), C_0_-C_3_ alkylbenzofurans and a dioxane.

In summary, there were several classes of compounds with an aromatic character (MAH, PAH, PHEN, benzofurans), others with a cyclic aliphatic character (alicyclic compounds, carbohydrate products) and compounds with an acyclic structure (linear alkanes and alkenes, isoprenoid alkanes and alkenes, and pentanoic acid derivatives). Remaining compounds contained atoms such as halogen, sulfur or nitrogen. Comparing the pyrolysis products of DOM in the study with previous reports, some similar features were found, although the DOM samples derived from different origins. For example, the predominant materials of CARB in our samples were consistent with the analysis results of DOM in soil in the Three Gorges Reservoir areas [13], and also in agreement with river DOM in the natural reserve of the Harz Mountains [25]. However, some differences can be also seen in the percentages. Obvious differences can be found between NCOMP in our study and above-mentioned samples. The contents of NCOMP were much lower than that in their samples. This difference is strong evidence of the Salt Lake brine being of less microbial DOM.

## 4. Conclusions

In this study, the abundance, compositions and variations of DOM in the evaporation process in solar ponds were investigated. The brine DOM samples were primarily composed of aliphatic groups, carbohydrates and aromatic groups, whereas the changes of DOM compositions were varied during solar pond exposure treatment. With increasing sunlight exposure time, the concentrations of DOC in the solar pond increased sharply. Further quantitative analysis indicated that the relative abundance of aliphatic increased significantly with increasing exposure time, whereas the reverse trends were observed for carboxylic acids, aromatics and carbohydrates. The pyrolysis products revealed that most DOM samples were aliphatic (methylene chain, terpenoid-like) and MAHs structures, possibly prevalently from bacterial organisms, but subject to considerable decays anyway. The presence of some sulfur-containing organics could indicate an anaerobic decay environment, but in the current system, microbiological processes were probably subordinate to the photo-induced DOM transformation. Aliphatics may be the main target compounds to be removed or treated for the extraction of high-value Salt Lake productions. The results provide a baseline dataset for DOM treatment or removal to guide the development of treatment strategies for the brine.

## Figures and Tables

**Figure 1 ijerph-19-09067-f001:**
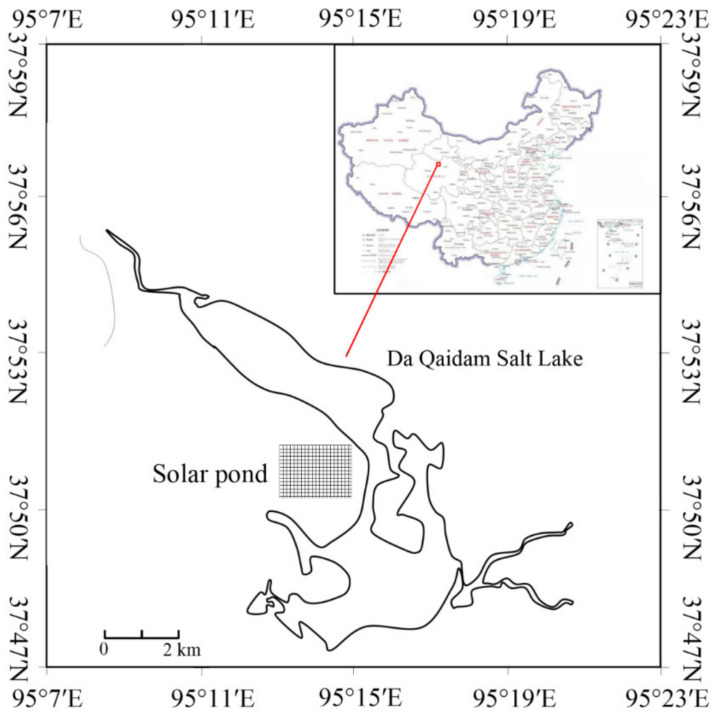
The map of the study area.

**Figure 2 ijerph-19-09067-f002:**
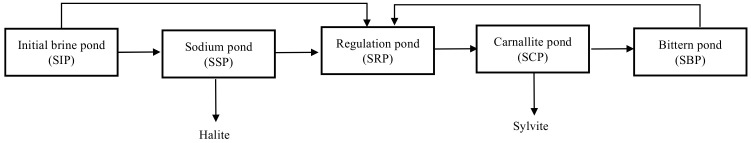
Scheme for the solar pond from Salt Lake brine.

**Figure 3 ijerph-19-09067-f003:**
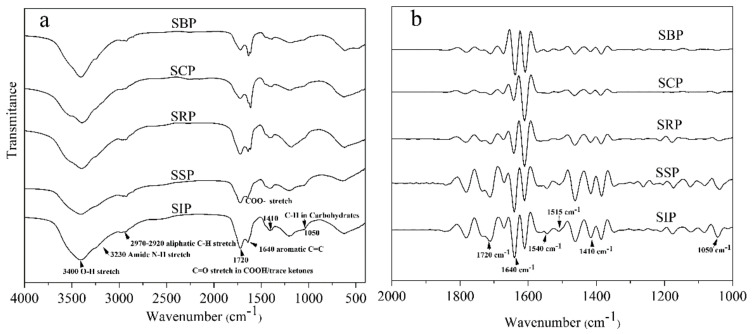
The FTIR spectra of DOM samples isolated from the solar pond process (**a**), and their corresponding second derivative spectra (**b**).

**Figure 4 ijerph-19-09067-f004:**
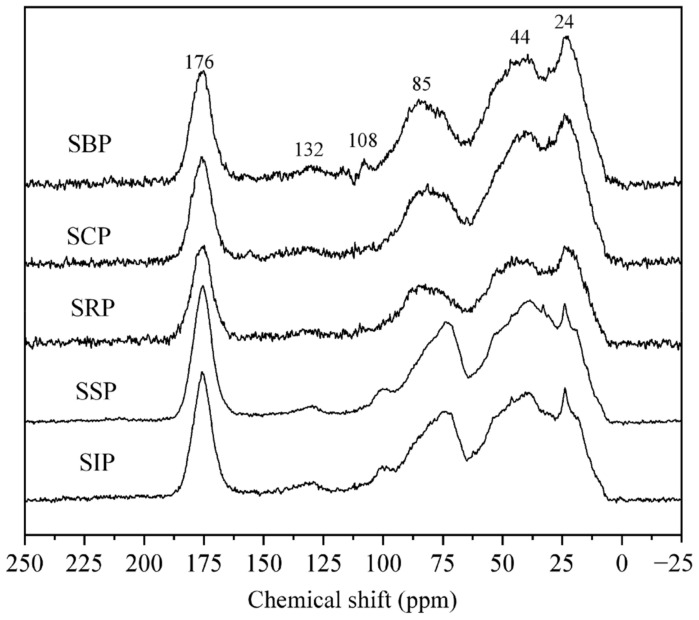
^13^C NMR spectra of DOM samples.

**Table 1 ijerph-19-09067-t001:** Variations in physical properties and chemical compositions of the studied Salt Lake brine samples in the solar pond process.

Sample	pH	TDS (g/L)	DIC (mg/L)	DOC (mg/L)	TN (mg/L)	Major Ions (g/L)						
						Cl^−^	SO_4_^2−^	B_2_O_3_	K^+^	Mg^2+^	Na^+^	Li^+^
SIP	7.4	273.5	33.6	23.4	6.6	107.8	26.2	0.6	4.1	15.0	119.8	0.1
SSP	7.2	290.6	40.9	28.1	7.0	110.5	36.7	3.0	4.8	18.4	117.1	0.1
SRP	6.6	326.9	16.4	68.8	18.1	128.9	60.3	7.4	11.7	47.2	71.1	0.3
SCP	4.6	353.5	8.3	198.3	45.1	180.6	12.1	18.3	0.8	108.3	31.9	1.5
SBP	4.6	479.6	18.5	330.8	85.5	277.1	18.6	40.1	0.5	114.9	25.5	3.0

**Table 2 ijerph-19-09067-t002:** Elemental composition, atomic ratios and stable isotopic analysis of DOM samples.

Sample	Mass%				Atomic Ratio			δ^13^C‰
	C%	H%	N%	O%	H/C	C/N	O/C	
SIP	41.6	3.8	2.1	52.4	1.1	22.7	1.0	−21.8
SSP	42.6	4.2	2.4	50.8	1.2	20.9	0.9	−21.0
SRP	37.1	4.7	1.9	56.2	1.5	22.4	1.1	−21.7
SCP	53.5	5.1	2.3	39.2	1.1	27.6	0.6	−21.4
SBP	47.5	5.2	1.8	45.5	1.3	30.1	0.7	−22.2

**Table 3 ijerph-19-09067-t003:** The relative carbon percentage of DOM functional groups and compound classes (total aliphatic and total aromatic) obtained from ^13^C NMR spectra.

Chemical Shift, ppm	0–45	45–65	Total	65–110	110–145	145–160	Total	160–190	190–220
Structural Category	Alkyl C	Methoxyl/N-alkyl C	Aliphatic	O-alkyl C	Aromatic C	Aromatic C–O	Aromatics	Carboxyl/Amide C	Ketones, Aldehydes
SIP	33.5	15.9	49.5	26.1	5.4	1.6	7.1	15.7	1.7
SSP	35.8	15.9	51.7	26.2	4.3	1.3	5.6	15.6	1.0
SRP	36.8	16.5	53.2	23.3	4.7	1.5	6.2	16.0	1.3
SCP	43.1	16.3	59.3	21.7	4.4	1.3	5.6	12.6	0.7
SBP	40.5	16.8	57.2	23.0	4.1	1.2	5.3	13.4	1.0

**Table 4 ijerph-19-09067-t004:** Pyrolysis product groups and relative proportions (%) of the DOM samples analyzed.

Categories	SIP	SSP	SRP	SCP	SBP
ALICYCL	25.81	24.99	27.39	28.08	25.62
CARB	26.88	24.87	27.59	21.60	19.92
MAH	16.79	17.52	17.61	20.05	22.54
MCC	11.57	15.69	11.21	11.51	11.22
NCOMP	1.66	1.50	1.43	1.60	1.84
OTHER	5.52	4.89	4.96	5.57	6.20
PAH	3.68	3.85	3.65	4.94	5.95
PHEN	3.65	3.49	3.28	3.50	3.88
SCOMP	2.19	0.76	1.18	0.70	0.66
HALOGENCOMP	1.65	0.68	0.47	1.26	1.13
PENT	0.32	0.77	0.48	0.40	0.32
LIG	0.30	0.40	0.36	0.43	0.32

Abbreviations: ALICYCL = alicyclic compounds, CARB = carbohydrate, MAH = monocyclic aromatic hydrocarbon, MCC = methylene chain compound, NCOMP = nitrogen-containing compound, PAH = polycyclic aromatic hydrocarbon, PHEN = phenols, SCOMP = sulfur-containing compound, PENT = pentanedioic acid, LIG = products of lignin and lignin-like phenolics.

## Data Availability

This published article includes all the data generated or analyzed during the study. The raw data of this research can be obtained by contacting the authors.

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
