# Peer review of "Characterization of Dissolved Organic Matter in Solar Ponds by Elemental Analysis, Infrared Spectroscopy, Nuclear Magnetic Resonance and Pyrolysis–GC–MS"

_ijerph, 2022, doi:10.3390/ijerph19159067_

Round 1
Reviewer 1 Report
The study area belongs to the plateau area, and the solar pool effect occurs significantly in the plateau area, and high-temperature brine is obtained at the same time. This study analyzed the chemical structure of DOM in brine in order to formulate an effective treatment strategy for DOM removal. At the same time, the chemical properties of dissolved organic matter also affect the existence and concentration of most pollutants in natural water, so it is very meaningful to understand the chemical structure of DOM. However, it is worth noting that this study is a common research in the field of environmental science and engineering. At the same time, there are many common problems in the manuscript, and I think it is still a long way from publication.
Q1: The abstract needs to be improved. It should not list the conclusions one by one, and it is suggested to strengthen the logic between the points.
Q2: What does "total DOM pool" mean?
Q3: What does “elemental analysis, Fourier transform infrared spectroscopy (FTIR) and nuclear mag- netic resonance (NMR) can only real the compositions of bulk or functional groups of DOM” mean?
Q4: In the introduction, "complementary chemical characterization techniques" (FTIR), (NMR), (py GC MS) are mentioned. Please explain in detail how these methods complement each other in the introduction.
Q5: The study area belongs to the plateau area. The solar pool effect takes place significantly in the plateau area, and high-temperature brine is obtained at the same time. What is the impact of this on the chemical composition of dissolved organic matter and the process of impact? It is suggested that the author improve this aspect into the introduction.
Q6: The title of the manuscript is " Changes in chemical composition of dissolved organic matter in the process of mineral extraction from salt lake brine ", but the content focuses more on chemical analysis methods. It is suggested to change the title and focus on methods.
Q7: Please explain that "this technology is powerless to explore the polar functional groups (-nh, -oh, -cooh) common in DOM in water." why did this study choose this method for analysis?
Q8: Please explain what is the purpose of this study to use complementary chemical characterization techniques to analyze DOM through method integration? What are the new findings of this integrated approach? Because these methods are common in analytical chemistry.
Q9: Please check in 2.2 Add quality control and instrument debugging parameters in the process of experimental analysis to physical and chemical characterization.
Q10: Please check in 2.3 Isolation and purification of DOM adds quality control and instrument debugging parameters in the process of experimental analysis.
Q11: Please add quality control and instrument debugging parameters in the process of experimental analysis in 2.5, 2.6 and 2.7.
Q12: The content of the conclusion needs to be improved. It is suggested to combine with practice to further prove the significance and importance of this study.
Q13: The grammar of the manuscript needs to be strengthened. Please provide language services for the manuscript.
Author Response
Response to Reviewer 1:
Comments and Suggestions for Authors
The study area belongs to the plateau area, and the solar pool effect occurs significantly in the plateau area, and high-temperature brine is obtained at the same time. This study analyzed the chemical structure of DOM in brine in order to formulate an effective treatment strategy for DOM removal. At the same time, the chemical properties of dissolved organic matter also affect the existence and concentration of most pollutants in natural water, so it is very meaningful to understand the chemical structure of DOM. However, it is worth noting that this study is a common research in the field of environmental science and engineering. At the same time, there are many common problems in the manuscript, and I think it is still a long way from publication.
Response: Thank you for your time and effort in the revision of our work, and your insightful comments.
Q1: The abstract needs to be improved. It should not list the conclusions one by one, and it is suggested to strengthen the logic between the points.
Response: The abstract has been modified, the conclusions were not listed one by one in the revised manuscript, and we believe the logic between the points have been strengthen, as follows: “Abstract: The abundance and chemical composition of dissolved organic matter (DOM) in the brine of solar ponds affect the efficiency of mineral extraction and evaporation rates of the brine, and cause undesired odor and color of the products. Here we report an investigation on the composition and changes of DOM in solar ponds from salt lake brine with multiple complementary analysis techniques. The results showed that the DOM deriving from salt lake brine was primarily composed by carbohydrates, aliphatic and aromatic compounds. The concentrations of dissolved organic carbon in solar ponds increased with exposure time up to 15−fold (from 23.4 to 330.8 mg/L) upon evaporation/irradiation of salt lake brine. Further qualitative analyses suggested that the relative abundance of aliphatic compounds (including functionalized ones) increased from 49.5% to 59.2% in solar pond process, while a generally opposite phenomenon was observed for carboxylic acid moieties, aromatics and carbohydrates, which decreased from 15.7%, 7.1% and 26.1% to 13.4%, 5.3% and 23.0%, respectively. The Pyrolysis−gas chromatography−mass spectrometry results revealed that the presence of some sulfur-containing organics implied some anaerobic biotic decays, but microbiological processes were probably subordinate to photo-induced DOM transformations. In the salt lake brine, exposure-driven decay decreased the abundance of polysaccharides and increased that of mono- and polyaromatic pyrolysis products. Our results here provide new insights for better understanding the changes of DOM chemical composition in the solar ponds of salt lake brine.”.
Q2: What does "total DOM pool" mean?
Response: It means all the components consisted of DOM, including chromophoric DOM, fluorescent DOM, and non-chromophoric/fluorescent DOM, and so on.
Q3: What does “elemental analysis, Fourier transform infrared spectroscopy (FTIR) and nuclear mag- netic resonance (NMR) can only real the compositions of bulk or functional groups of DOM” mean?
Response: It means above mentioned methods can not directly identify the presence of specific
compounds at molecular scale, such as molecular structural formula of individual component.
Q4: In the introduction, "complementary chemical characterization techniques" (FTIR), (NMR), (py GC MS) are mentioned. Please explain in detail how these methods complement each other in the introduction.
Response: We have added more contents to explain these methods complement each other in the introduction, “For example, elemental analysis can provide the DOM elemental composition (C, H, O, N, S), while FTIR can further show the functional groups of DOM. In addition, NMR not only provides more detailed functional structures, but also provides their quantitative information. Recent studies have shown that the technique of Pyrolysis−gas chromatography−mass spectrometry (Py−GC−MS) has extended the characterization level to the molecular scale, which can break macromolecular DOM into fragments and enable rapid screening of macromolecular constituents of DOM, such as protein, lignin, carbohydrates and so on [23-26]”.
Q5: The study area belongs to the plateau area. The solar pool effect takes place significantly in the plateau area, and high-temperature brine is obtained at the same time. What is the impact of this on the chemical composition of dissolved organic matter and the process of impact? It is suggested that the author improve this aspect into the introduction.
Response: We have added the contents to state the impact of the plateau area and high-temperature brine on the chemical composition of DOM and the process of impact, “Moreover, the abundance and compositions of DOM will undergo significant changes, such as photodegradation and evaporation loss due to intense sunlight exposure and high-temperature in solar ponds, which will further affect the degree of evaporation and the quality of the products extracted from the brine. The investigations on the changes in DOM composition in the solar ponds will provide a baseline dataset from which to understand future changes during exploitation of salt lake resources.”.
Q6: The title of the manuscript is " Changes in chemical composition of dissolved organic matter in the process of mineral extraction from salt lake brine ", but the content focuses more on chemical analysis methods. It is suggested to change the title and focus on methods.
Response: The original title of the manuscript has been changed to “Characterization of dissolved organic matter in solar ponds by elemental analysis, infrared spectroscopy, nuclear magnetic resonance, pyrolysis-GC-MS”.
Q7: Please explain that "this technology is powerless to explore the polar functional groups (-nh, -oh, -cooh) common in DOM in water." why did this study choose this method for analysis?
Response: Although not ideal, Py-GC-MS are probably the best techniques used so far for structural investigations of macromolecular organic materials. Moreover, FTIR and NMR can provide more information on the polar functional groups, which can fill this gap. Therefore, we stated in the text “Thus, the complementary chemical characterization techniques were essential for a comprehensive understanding of the chemical composition of DOM.”
Q8: Please explain what is the purpose of this study to use complementary chemical characterization techniques to analyze DOM through method integration? What are the new findings of this integrated approach? Because these methods are common in analytical chemistry.
Response: Due to the complex and heterogeneous, there is no single method can reflect the DOM composition. Thus, characterization of DOM using different method was popular by researchers in the field of environment and geochemistry to investigate the distribution, sources and composition of DOM derived from different origins. The purpose of this study was to provide new insights into the compositions and changes for DOM transformation in the solar pond through method integration. Although these methods are common in analytical chemistry, the information on the composition and changes of DOM was what we tend to focus on.
Q9: Please check in 2.2 Add quality control and instrument debugging parameters in the process of experimental analysis to physical and chemical characterization.
Response: We have added quality control and instrument debugging parameters for the analyzed methods. For example “...... Each sample was diluted (1:10) with ultrapure water prior to analysis due to hypersaline characteristics. All the values were obtained with the variation coefficients less than 3.0%. and the detection limits were 0.1 ppm......”.
Q10: Please check in 2.3 Isolation and purification of DOM adds quality control and instrument debugging parameters in the process of experimental analysis.
Response: The quality control and instrument debugging parameters have been added in the process of experimental analysis.
Q11: Please add quality control and instrument debugging parameters in the process of experimental analysis in 2.5, 2.6 and 2.7.
Response: The quality control and instrument debugging parameters have been added in the process of experimental analysis.
Q12: The content of the conclusion needs to be improved. It is suggested to combine with practice to further prove the significance and importance of this study.
Response: The content of the conclusion has been modified by us and the significance and importance of this study were also be highlighted as follows “In this study, the abundance, composition and variations of DOM with evaporation process in solar ponds were investigated. The brine DOM samples were primarily composed of aliphatic groups, carbohydrates and aromatic groups, whereas the changes of DOM composition were varied during solar pond exposure treatment. With increasing solar exposure time, the concentrations of DOC in the solar pond increased sharply. Further quantitative analysis indicated that the relative abundance of aliphatic increased significantly with increasing exposure time, whereas the reverse trends were observed for carboxylic acids, aromatics and carbohydrates. The pyrolysis products revealed that most DOM samples were aliphatic (methylene chain, terpenoid-like) and MAHs structures, possibly prevalently from bacterial organisms but subject to considerable decays anyway. The presence of some sulfur-containing organics could indicate an anaerobic decay environment, but in the current system microbiological processes were probably subordinate to the photo-induced DOM transformation. Aliphatics may be the mainly target compounds to remove or treat for the extraction of high-value salt lake productions. The results provide a baseline dataset for DOM treatment or removal to guide development of treatment strategies for the brine.”.
Q13: The grammar of the manuscript needs to be strengthened. Please provide language services for the manuscript.
Response: The manuscript has been checked and polished by a few cooperators with good English writing skill, we believe it can be accepted now.
Reviewer 2 Report
review attached

Author Response
Response to Reviewer 2:
General comments
The DOM quality was measured in salt lake brine of different process state. Complementary
DOM characterizing methods were performed. The authors received a good data basis for
discussion. Especially the Pyr-GC/MS results are provided in very comprehensive way. The
detail discussions of the results are so far well structured and understandable and of good
quality. The summarizing statements in the abstract or in the conclusions require more precise
chemical phrasing to avoid misunderstandings. Some more recent literature can be integrated
for discussion of DOM processing.
Response: Thank you for your time and effort in the revision of our work, and your insightful comments.
Detail comments
Abstract, line 20-23: aliphatics increased and carboxylic acids and carbohydrates the
opposite?
The problem is, carboxylic acids can be also relative aliphatic like saturated fatty acids.
Carbohydrates are also relative aliphatic. So what is meant with aliphatics here?
Hydrocarbons, aliphatic alcohols?
Response: The results were obtained from FTIR (section 3.3) and 13C NMR (section 3.4) analysis, and the structural categories of organic C or functional groups have been described in detail elsewhere (Hertkorn et al., 2013; Zhang et al., 2014). Due to the complex and heterogeneous, DOM was usually classified based on their general groups and features, rather than individual component.
Lines 24-255: What dominated really? Terpenoid, carbohydrate and aromatics stands nearly
for the entire possible chemistry. Can this be specified somewhat more in detail?
Response: As is mentioned above, DOM is complex and heterogeneous which can be classified different categories based on their general groups and features using different methods. For example, the 13C NMR spectra (section 3.4) can be divided into seven main regions, i.e., 0-45 ppm (alkyl C), 45-65 ppm (methoxyl/N-alkyl C), 65-110 ppm (O-alkyl C), 110-145 ppm (aromatic C); Py-GC-MS can give more detail information on DOM structures, such as alkane/alkene pairs, alkyl-substituted monoaromatic and polyaromatic hydrocarbons (MAHs/PAHs) and alicyclic compounds. However, more knowledge on individual compound subject to advanced analysis technology nowadays.
Line 45-68: Here a good overview can be found which analytical methods exist and what can
they perform addressing DOM quality. However the method with the up to date highest
resolution addressing DOM quality, FTICR-MS was not mentioned here, although since 2002
there were published hundreds of contributions addressing DOM quality and DOM
transformations.
Response: We have added the content refer to FTICR-MS, “Fourier Transform Ion Cyclotron Resonance Mass Spectrometry (FT-ICR-MS) is characterized by a high degree of mass accuracy and precision, and can give distributions of elemental formulae. However, these studies are limited to high-purity DOM fractions and expensive costs.”.
Line 63, here I miss a very important reference for SPE with PPL:
Dittmar, T.; Koch, B.; Hertkorn, N.; Kattner, G., A simple and efficient method for the solidphase extraction of dissolved organic matter (SPE-DOM) from seawater. Limnology and
Oceanography-Methods 2008, 6, 230-235
Response: The reference Dittmar et al., (2008) has been cited now, “29. Dittmar, T.; Koch, B.; Hertkorn, N.; Kattner, G. A simple and efficient method for the solid-phase extraction of dissolved organic matter (SPE-DOM) from seawater. Limnol. Oceanogr-Meth. 2008, 6, 230-235.”
Section 2.7, pyrolysis GC/MS: No derivatization was performed (for example with TMAH in
MeOH)? How can molecules with polar groups be detected (-COOH, -NH2…)?
Response:Thank you for your comment. Both Pyrolysis GC/MS and TMAH GC/MS involve a thermal pulse that is used to break macromolecular DOM into fragments that are amenable to GC/MS to obtain a rapid screening of macro-molecular constituents of DOM. Although THM GC/MS is less prone to secondary rearrangements due to protection of functional groups and provides more detailed information on the polar groups, Pyrolysis GC/MS has an advantage over THM GC/MS in that it is more informative on microbial DOM components. Of course, it is true that the Pyrolysis GC/MS performed poorly when it probe into DOM with polar functional groups (-NH, -OH, -COOH).
Line 159: Is the DC relevant at all to discuss? The DOC and DIC results are available, so
what sense makes discussion of DC?
Response: DOC were calculated from the difference between DC and DIC. DC values herein confirmed the accuracy of DOC concentration, there is no much sense. Thus, we have deleted the discussion of DC from the original manuscript.
Line 183-184: I disagree, the so called aliphatics in DOM do mainly contain O and / or N,
otherwise they would be hydrocarbons which would not show high solubility. Not all
carbohydrates do contain N. I suggest more careful chemical phrasing.
Response: Thank you for your comment. The original sentences have been modified by “As depicted in Table 2, the DOM samples in initial stages of the solar pond generally contained lower values of H/C and C/N than that of late stages of the solar ponds, while O/C ratios exhibited an opposite trend. The results indicated that the contents of saturated aliphatic increased but N/O-containing compounds decreased with increasing exposure time in the solar pond process.”.
Line 236: the benzene region ppm 132, the authors should make a statement here, is as small
as expected or other evaluation?
Response: We have added the content to evaluate the result, “......accounted for 4.1-5.4%, which was generally lower than that in estuary in the eastern coast of Hainan Island (3.97-15.96%) [42].”.
Reviewer 3 Report
Dear authors,
Your article contains a lot of interesting data. However, their interpretation must be significantly better. Some comments are included in the attached pdf.
Best regards.

Author Response
Response to Reviewer 3:
Comments and Suggestions for Authors
Dear authors,
Your article contains a lot of interesting data. However, their interpretation must be significantly better. Some comments are included in the attached pdf.
Best regards.
Response: Thank you for your time and effort in the revision of our work, and your insightful comments.
- Throughout the text, you cite relatively old publications. However, I believe newer publications with similar topics have been published, which would be appropriate to include in the text.
Response: Thank you for your comment. We have added more newer publications in the text and the relatively old publications have been removed. For example, the added newer publications:
Zhao, L.; Gao, L.; Guo, L. Seasonal Variations in Molecular Size of Chromophoric Dissolved Organic Matter From the Lower Changjiang (Yangtze) River. J. Geophys. Res-Biogeo. 2021, 126, 1-17.
O'Connor, J. A.; Lu, K.; Guo, L.; Rosenheim, B. E.; Liu, Z. Composition and lability of riverine dissolved organic matter: Insights from thermal slicing ramped pyrolysis GC–MS, amino acid, and stable isotope analyses. Org. Geochem. 2020, 149, 104100.
Gerea, M.; Pérez, G. L.; Unrein, F.; Soto Cárdenas, C.; Morris, D.; Queimaliños, C. CDOM and the underwater light climate in two shallow North Patagonian lakes: evaluating the effects on nano and microphytoplankton community structure. Aquat. Sci. 2016, 79, 231-248.
Li, Y.; Harir, M.; Uhl, J.; Kanawati, B.; Lucio, M.; Smirnov, K. S.; Koch, B. P.; Schmitt-Kopplin, P.; Hertkorn, N. How representative are dissolved organic matter (DOM) extracts? A comprehensive study of sorbent selectivity for DOM isolation. Water Res. 2017, 116, 316-323.
Hertkorn N, Harir M, Koch B P, Michalke B, Schmitt-Kopplin P. 2013. High-field NMR spectroscopy and FTICR mass spectrometry: powerful discovery tools for the molecular level characterization of marine dissolved organic matter. Biogeosciences, 10(3): 1583-1624.
The removed publications following as:
Helms, J. R.; Stubbins, A.; Ritchie, J. D.; Minor, E. C.; Kieber, D. J.; Mopper, K. Absorption spectral slopes and slope ratios as indicators of molecular weight, source, and photobleaching of chromophoric dissolved organic matter. Limnol. Oceanogr. 2008, 53, 955-969.
Wang, X.; Goual, L.; Colberg, P. J. Characterization and treatment of dissolved organic matter from oilfield produced waters. J. Hazard. Mater. 2012, 217-218, 164-170.
Zhang, Y.; Liu, M.; Qin, B.; Feng, S. Photochemical degradation of chromophoric-dissolved organic matter exposed to simulated UV-B and natural solar radiation. Hydrobiologia 2009, 627, 159-168.
Kim, S.; Simpson, A. J.; Kujawinski, E. B.; Freitas, M. A.; Hatcher, P. G. High resolution electrospray ionization mass spectrometry and 2D solution NMR for the analysis of DOM extracted by C18 solid phase disk. Org. Geochem. 2003, 34, 1325-1335.
Aiken, G. R.; McKnight, D. M.; Thorn, K.; Thurman, E. Isolation of hydrophilic organic acids from water using nonionic macroporous resins. Org. Geochem. 1992, 18, 567-573.
- Abstract: Be more specific in the Abstract, and state the most important findings (express them numerically).
Response: Thank you for your comment. We have added more detailed information to state the important findings. For example, “The concentrations of dissolved organic carbon in solar ponds increased with exposure time up to 15−fold (from 23.4 to 330.8 mg/L) upon evaporation/irradiation of salt lake brine. Further qualitative analyses suggested that the relative abundance of aliphatic compounds (including functionalized ones) increased from 49.5% to 59.2% in solar pond process, while a generally opposite phenomenon was observed for carboxylic acid moieties, aromatics and carbohydrates, which decreased from 15.7%, 7.1% and 26.1% to 13.4%, 5.3% and 23.0%, respectively”.
- Line 31: Words from the title should not be repeated.
Response: The Keywords have been changed to “Chemical composition; Salt lake brine; Evaporation; Mineral extraction; Transformations”.
- Introduction: This section summarizes important information, but only in very general terms. There is much scope for improvement in this chapter.
Response: Thank you for your comment, we have added more content to improve this chapter. For example, “ For example, elemental analysis can provide the DOM elemental composition (C, H, O, N, S), while FTIR can further show the functional groups of DOM. In addition, NMR not only provides more detailed functional structures, but also provides their quantitative information. Recent studies have shown that the technique of Pyrolysis−gas chromatography−mass spectrometry (Py−GC−MS) has extended the characterization level to the molecular scale, which can break macromolecular DOM into fragments and enable rapid screening of macromolecular constituents of DOM, such as protein, lignin, carbohydrates and so on [23-26]. However, this technique is powerless when it probe into DOM with polar functional groups (-NH, -OH, -COOH) typically present in aquatic DOM [23]. Fourier Transform Ion Cyclotron Resonance Mass Spectrometry (FT-ICR-MS) is characterized by a high degree of mass accuracy and precision, and can give distributions of elemental formulae. However, these studies are limited to high-purity DOM fractions and expensive costs”.
- Line 68: What was the goal of the study?
Response: The main goal of the study was stated in the text, “This work will complement and extend our understanding of the chemical structures and changes of DOM in solar ponds, which is essential to guide development of environmental management and treatment strategies for high-value productions extracted from salt lakes.”.
- Line 74: Is it area or volume?
Response: Thank you for your comment. It is the area and original writing was false, the original sentence has been modified by “It has an area of 240 km2,......”.
- Line 75: Is it correct?
Response: Yes, it is corrected, the depth of salt lakes are generally low.
- Line 89: Better describe the sampling procedure
Response: We added the content to describe the sampling procedure, “Before collection, each of the acid cleaned carboys were thoroughly rinsed with the samples.”.
- It is necessary to rework this chapter. It contains many exciting data, but their interpretation is not reasonable. Describe the results, so they are understandable even for readers who are not experts in this field.Discuss more with the literature.
Response: Thank you for your comment. We have added more contents to describe the data as well as the results. For example, in section 3.1: “ The pattern of the inorganic constituents was similar to that of water produced by shale gas and tight gas sands from certain oil wells in central and western U.S., which were characterized by high magnesium concentration [31].”; In section 3.2: “As depicted in Table 2, the DOM samples in initial stages of the solar pond generally contained lower values of H/C and C/N than that of late stages of the solar ponds, while O/C ratios exhibited an opposite trend. The results indicated that the contents of saturated aliphatic increased but N/O-containing compounds decreased with increasing exposure time in the solar pond process.”; In section 3.4:“...... accounted for 4.1-5.4%, which was generally lower than that in estuary in the eastern coast of Hainan Island (3.97-15.96%) [41].”, “In general, the components with functional groups were more likely to decay or transform compared to saturated hydrocarbons in the solar pond. Thus, the aliphatics may be the target compounds to remove or treat for the extraction of high-value salt lake productions.”.
- Highlight the significance of your findings more.
Response: Thank you for your comment. We have added more contents to highlight the significance of our findings. For example, In section 3.4, “In general, the components with functional groups were more likely to decay or transform compared to saturated hydrocarbons in the solar pond. Thus, the aliphatics may be the target compounds to remove or treat for the extraction of high-value salt lake productions.”. In section 4, “In this study, the abundance, composition and variations of DOM with evaporation process in solar ponds were investigated. The brine DOM samples were primarily composed of aliphatic groups, carbohydrates and aromatic groups, whereas the changes of DOM composition were varied during solar pond exposure treatment. With increasing solar exposure time, the concentrations of DOC in the solar pond increased sharply. Further quantitative analysis indicated that the relative abundance of aliphatic increased significantly with increasing exposure time, whereas the reverse trends were observed for carboxylic acids, aromatics and carbohydrates. The pyrolysis products revealed that most DOM samples were aliphatic (methylene chain, terpenoid-like) and MAHs structures, possibly prevalently from bacterial organisms but subject to considerable decays anyway. The presence of some sulfur-containing organics could indicate an anaerobic decay environment, but in the current system microbiological processes were probably subordinate to the photo-induced DOM transformation. Aliphatics may be the mainly target compounds to remove or treat for the extraction of high-value salt lake productions. The results provide a baseline dataset for DOM treatment or removal to guide development of treatment strategies for the brine.”.
Round 2
Reviewer 1 Report
The manuscript has been well revised, and the rearrangement of the introduction and abstract looks very good. The comments of reviewers were answered one by one.
Author Response
Thank you very much!
Reviewer 2 Report
Line 62: The important DOM method FTICR-MS now was mentioned as recommended, however a reference to it is missing.
There is no response to the comment addressing line 294
Author Response
Response to Reviewer 2:
Comments and Suggestions for Authors
Line 62: The important DOM method FTICR-MS now was mentioned as recommended, however a reference to it is missing.
Response: Two references mentioned FTICR MS have been added:
- Li, H.; Minor, E. C. Dissolved organic matter in Lake Superior: insights into the effects of extraction methods on chemical composition. Environ. Sci-Proce Imp. 2015, 17, 1829-1840.
- Xiang, Y. Y.; Gonsior, M.; Schmitt-Kopplin, P.; Shang, C. Influence of the UV/H2O2 Advanced Oxidation Process on Dissolved Organic Matter and the Connection between Elemental Composition and Disinfection Byproduct Formation. Environ. Sci. Technol. 2020, 54, 14964-14973.
There is no response to the comment addressing line 294
Response: We are very sorry for no response to the comment addressing line 294. We have added the contents and references here for disscussion, “In previous studies, the aromatics have been confirmed being of the preferential photo-degradation of DOM components [48, 49]. The increased trend of MAHs here may be related to the decomposition of the polyphenolic and condensed aromatic compounds, which released more low molecular weight MAHs [50].”.
Reviewer 3 Report
Dear authors,
Thanks for improving the article. Its quality has increased significantly. However, I can recommend improving the discussion.
Best regards.
Author Response
Response to Reviewer 3:
Comments and Suggestions for Authors
Dear authors,
Thanks for improving the article. Its quality has increased significantly. However, I can recommend improving the discussion.
Response: Thank you for your comment. We have added more contents to improving the discussion. For example:
In section 3.2 “......The phenomenon suggested the transformations of unsaturated hydrocarbons to saturated aliphatic compounds in solar ponds.”;
In section 3.3 “......which were conformable to the the results of elemental analysis.”
In section 3.5 1)“ ......In general, the percentanges of ALICYCL exhibitaed decreasing trend, which was in agreement with the elemental composition and 13C NMR results.”; 2)“The decreased tends of CARB exposure to sunlight in solar ponds further enhanced this conclusion.”; 3) In previous studies, the aromatics have been confirmed being of the preferential photo-degradation of DOM components [48, 49]. The increased trend of MAHs here may be related to the decomposition of the polyphenolic and condensed aromatic compounds, which released more low molecular weight MAHs [50].”; 4) “......which is an important marker of vascular plants, mainly deriving from terrigenous sources. The low percentage (0.30-0.43%) indicated autochthonous characteristics of these DOM. According to carbon stable isotopic analysis, this materials exhibited significantly in situ origins, and our results here appear to reinforce this viewpoint.”; 5) “Compared the pyrolysis products of DOM in the studied with previous reports, some similar features were found although the DOM samples derived from different origins. For example, the predominant materials of CARB in our samples was consistent with the analysis results of DOM in soil in the Three Gorges Reservoir areas [13], and also in agreement with river DOM in the natural reserve of the Harz Mountains [25]. However, some differences can be also seen from the percentages. Obvious differences can be found between NCOMP in our study and above mentioned samples. The contents of NCOMP were much lower than that in their samples. This difference are strong evidence of the salt lake brine being of less microbial DOM.”.